# Gait assessment using a 2D video-based motion analysis app in healthy subjects and subjects with lower limb amputation – A pilot study

**Frithjof Doerks**[1], **Fenna Harms**[1], **Michael Schwarze**[1,2], **Eike Jakubowitz**[1], **Bastian Welke**[1]*

**1** Hannover Medical School, Department of Orthopaedic Surgery, DIAKOVERE Annastift, Laboratory for Biomechanics and Biomaterials, Hannover, Germany, **2** Department for Medical Technology, Bremerhaven University of Applied Sciences, Bremerhaven, Germany

* welke.bastian@mh-hannover.de

## Abstract

### Introduction

Although three-dimensional marker-based motion analysis is the gold standard for biomechanical investigations, it is time-consuming and cost-intensive. The conjunction of monocular video recordings with pose estimation algorithms addresses this gap. With the Orthelligent VISION app (OPED GmbH) a commercial and easy-to-use tool is now available for implementation in everyday clinical practice. The study investigates the accuracy of the 2D video-based system in measuring joint kinematics, expressed as range of motion, compared to an optoelectronic 3D motion analysis system as the gold standard.

### Materials and methods

Its accuracy was determined by synchronously measuring ten healthy subjects with Orthelligent and the optoelectronic 3D motion analysis system Qualisys (Qualisys AB) during level walking and at different treadmill walking speeds (1 m/s; 1.4 m/s; 1.8 m/s). Range of motion (RoM) of lower limb joints and time-distance parameters were compared using Bland-Altman plots, t-tests, and correlations between systems. Kinematic outputs of two subjects with a lower limb amputation were also analyzed.

### Results

The mean RoM deviation was smaller for the knee (3.8°) and hip joints (3.7°) than for the ankle joint (5.4°), but differed significantly between systems in most conditions. The correlation range was $0.36 \leq r \leq 0.83$, with best results for 1 m/s treadmill walking (mean $r = 0.71$ across joints). While the accuracy was affected by high inter-subject variability, individual RoM changes from slow to fast walking did not differ between the systems. The kinematics of the prosthetic and sound leg of individuals with an

**Data availability statement:** The raw data from all the subjects' trials from the gait analysis are uploaded as supporting material. An explanation of the data can be found in the Supporting Information.

**Funding:** Publication costs are covered by the German Research Foundation (DFG) and the Open Access Publication Fund of Hannover Medical School (MHH). The funders had no role in study design, data collection and analysis, decision to publish, or preparation of the manuscript.

**Competing interests:** The authors have declared that no competing interests exist.

amputation exhibited characteristic patterns in the video-based system, even though side differences were smaller compared to the optoelectronic measurement.

## Conclusions

The rather high inter-subject variability would make future comparisons between individuals challenging. Nonetheless, the app shows potential for intra-subject progress monitoring.

---

## Introduction

Motion capture systems are a widely used method for quantifying human movement in sports science, ergonomics, industry, and clinical applications [1–4]. While optical, three-dimensional, marker-based motion capture (3D-MB) in the laboratory is the gold standard for biomechanical investigations, Mündermann et al. [5] advocated implementing novel markerless techniques more than 15 years ago. Such systems are easy to use, time-efficient, and potentially more relevant in clinical applications as they can be used outside the laboratory [5]. Nowadays, various markerless methods are used for clinical measurements [6]. Scataglini et al. [7] summarized that such methods show good to excellent results in terms of time-distance parameter, moderate to excellent agreement in sagittal kinematics of the hip and knee joint, and low validity and reliability in ankle kinematics when compared with marker-based methods. Commercially available systems such as Theia3D show good results for both healthy subjects and clinical patients, but also require equipment with several cameras [8]. A current review shows that the most recent studies on markerless motion capture use camera or smartphone videos in conjunction with pose estimation algorithms for motion analysis [6]. The use of such everyday devices falls under the concept of monocular markerless motion capture, offering cost-efficiency and user-friendliness. Due to the progress of machine learning pose estimation models for 2D motion analysis in recent years, they enable objective gait and posture analysis in a clinical setting [9]. In addition to pure measurement, the algorithms now even have the potential to influence the decision-making process through automated screening, detection and diagnosis options for movement disorders [10]. However, few studies have compared the accuracy of joint angle estimation using marker-based methods [11].

Washabaugh et al. [12] compared four open-source pose estimation methods based on sagittal video recordings with a marker-based method in healthy subjects. They focused on the knee and hip joint kinematics and reported the lowest mean errors over the entire gait cycle with 3.7° ± 1.3° and 4.6° ± 1.8° for the hip joint and 5.1° ± 2.5° for the knee joint. Van Hooren et al. [13] reported root mean square error (RMSE) values between 5.1° and 10.9° for knee and hip joint angles at different running speeds when comparing two computer vision techniques with the gold standard approach. Temporal misalignments and vertical offsets were observed and corrected during the data analysis process [13]. Menychtas et al. [14] showed that

although the joint angles do not differ significantly, in some cases, when comparing two pose estimation algorithms with the marker-based measurement of older people, the curves showed high variability and a pattern that clearly deviated from the physiological pattern. Horsak et al. [15] investigated the validity of smartphone-based markerless motion analysis in healthy and simulated pathological gait. They found an overall RMSE of 5.8° and a peak error of 11.3° with the best agreement for healthy gait [15]. A very high correlation between the range of motion (RoM) determined with a smartphone and the gold standard and a high to very high correlation for the time-distance parameters were found by Hu et al. [16].

All these reported results are from studies where computer vision techniques were applied by professional researchers in a scientific setting. With the Orthelligent VISION app (Orthelligent), the company OPED GmbH (Valley, Germany) has launched such a 2D-video-based motion capture (2D-VB) tool as a CE-registered commercial product for use in muscu-loskeletal therapy and diagnostics. The app automatically generates a gait analysis report after capturing a 30 s digital video using a cloud-based proprietary solution without the need for further data processing. In contrast to open source solutions such as DeepLabCut, the data does not have to be labeled manually initially and does not have to be retrained for each recording scenario [17]. The manufacturer claims that the capturing can be simply integrated into everyday clinical practice in a time-efficient manner without the need for skilled personnel. As no separate laboratory is required, such a gait screening tool is now available to a wide range of users. Depending on the objective, whether in research or in clinical practice, sufficient accuracy of the measuring system must be guaranteed for a meaningful application. This study considers a deviation of 5° acceptable, based on the results of current 2D-VB measurements and the position that 2D-VB complements rather than competes with 3D-MB for efficient patient screening with minimal accuracy loss [18].

Up to now, the results of an investigation into the system's accuracy are only available as a whitepaper on the man-ufacturer's website [19]. A mean Pearson correlation coefficient of $r = 0.989$ is reported for the maximum value of knee joint flexion and extension of 15 subjects when compared with a goniometer, with a mean deviation of $d = 7.35°$ between systems [19]. Additional information regarding the subject cohort and methods used is missing. Since the output report of Orthelligent provides a complete kinematic analysis of a full gait cycle of the lower limbs, there is a high need for a sys-tem comparison with the gold standard motion capture technique and analysis of other joints. While standardized in-vitro examinations such as those conducted by Mihcin et al. [20] would be desirable, this is not possible with the computer vision-based solution for in-vivo applications, so the focus is on in-vivo gait analysis. As the RoM is a key parameter in clinical biomechanics [21–23], the main research question is: How accurate is the 2D-VB system in measuring joint kine-matics expressed by the RoM compared to the gold standard 3D-MB? Due to the continuous development and growing size of the training dataset, the primary hypothesis is that the knee and hip joint deviations are below the acceptable deviation of 5°, but at least below the value in the whitepaper (7.35°) [19]. Based on previous results [7], higher deviations are anticipated for the ankle. Considering the high variability of results reported for other 2D-VB systems [14] and in the whitepaper [19], we expect that the system will perform better for some subjects than others, resulting in high inter-subject variability but good intra-subject accuracy. Since Orthelligent is also intended for use in prosthetic and orthotic care and the kinematics of subjects with an amputation differ from healthy subjects due to the prosthetic fitting [24], cases with an amputation will also be investigated. The different leg surface shape in subjects with an amputation is expected to result in greater deviations between 3D-MB and 2D-VB systems than in subjects with a healthy gait.

## Materials and methods

### Participants

Ten healthy subjects (f = 8, 25.2 ± 2.8 years, 172.9 ± 7.2 m, 67.3 ± 12.9 kg) and two subjects with a lower limb amputation (data in Table 1) volunteered for this pilot study. Exclusion criteria for the healthy subjects were musculoskeletal impair-ments and no previous experience in walking on a treadmill. The subjects with an amputation had to have at least a mobil-ity grade 3 as defined by Greitemann [25], the ability to walk without the use gait aids, and no further musculoskeletal

**Table 1. Characteristics of the subjects with a lower limb amputation.**

| Subject | Level of amputation | Sex | Age (y) | Weight (kg) | Height (m) | Prosthetic fittings |
|---|---|---|---|---|---|---|
| TT | transtibial | m | 62 | 101.5 | 1.94 | 1C61 Triton VS (Otto Bock HealthCare Deutschland GmbH, Duderstadt, Germany) |
| TF | transfemoral | f | 65 | 67.3 | 1.60 | 1C63 Triton LP, C-Leg (Otto Bock HealthCare Deutschland GmbH) |

impairments beyond the unilateral amputation. The study was approved by the local ethics committee of the Hannover Medical School (#11394/24), and all subjects gave written informed consent before participating in the study. The participants were recruited between 2 May and 24 July 2024.

### Instrumentation and experimental protocol

First, anthropometric data were collected for each subject. Retroreflective markers were attached to each subject's skin according to the Plug-in-Gait model [26]. In a static trial, additional medial markers for the ankle and knee joints were used to determine joint widths. For the first part of the investigation, a treadmill (mercury med 150/50, h/p/cosmos, Germany) was placed in the center of the laboratory. A tablet (IPad Pro, 11 inches, 4th generation, Apple Inc., Cupertino, California, USA) with Orthelligent (Version 1.1.0–1.2.2, OPED GmbH, Germany) was set up on a tripod parallel to the gait track in accordance with the manufacturer's instructions. The healthy subjects were asked to walk on the treadmill at speeds of 1.0, 1.4, and 1.8 m/s for 30 s after a familiarization period of 30 s per condition (S1 Table). The speeds were the normal walking speed of young adults [27] as well as one below and one above without exceeding the walking to running transition speed [28]. In addition to the standardized conditions, all subjects were asked to walk barefoot up and down a 10 m walkway in the gait laboratory for 30 s at a self-selected walking speed. The capture interval was set to 30 s, as this is the requirement of the 2D-VB system for automatic report creation. The level walking condition was repeated three times (S1 Table). The subjects with an amputation only completed the level walking part for safety reasons (S2 Table).

Kinematics were acquired simultaneously with a 200 Hz motion capture system consisting of twelve infrared cameras (Miqus, Qualisys AB, Gothenburg, Sweden) and the Orthelligent VISION app.

### Data analysis

For 2D-VB, a report was automatically generated for each measurement, including mean sagittal kinematic joint angle curves of lower limbs and selected time-distance parameters. As the app does not provide an option to export numbers, data were extracted from digital graphs using a web-based extraction tool (WebPlotDigitizer, Automeris LLC, Austin, Texas, USA). The 3D-MB data was processed with Visual 3D (Vers. 2023.10.1, HAS-Motion, Kingston, Canada). The level walking trials were cut according to the viewing window of the tablet and gait events were automatically generated in all trials using a kinematic-based algorithm according to Zeni et al. [29]. Motion capture data was time-normalized to the gait cycle. The kinematic data and time-distance parameters for the three level walking trials were averaged separately for the left and right leg side with both systems and for the treadmill trials within the 30 s interval. The post-processing was done using a custom-written code in MATLAB (Vers. R2021b, The MathWorks, Inc., USA).

### Statistical analysis

Besides qualitative analysis of the time-normalized joint angle curves of both systems, Bland-Altman diagrams were created to graphically examine both systems and test the primary hypothesis [14,30]. Moreover, joint angle RoMs were calculated in accordance to Hu et al. [16], and validity was assessed using Pearson's correlation coefficient $r$. The evaluation of the correlation was based on the ranges proposed by Mukaka [31]: $r < 0.30$: negligible correlation, $0.30 \leq r < 0.50$:

low correlation, 0.50 ≤ r < 0.70: moderate correlation, 0.70 ≤ r < 0.90: high correlation, 0.90 ≤ r ≤ 1.00: very high correlation. Furthermore, a paired t-test with descriptive p-values was conducted to compare sagittal lower limb joint RoM and time-distance parameters measured with both systems for each condition. The intraclass correlation coefficient (ICC(3,1)) was calculated to assess the consistency between the two measurement systems. ICC values for consistency are reported in relation to the study of Koo and Li [32] in the following manner: poor (<0.50), moderate (0.50–0.75), good (0.75–0.90), and excellent (>0.90). All statistic calculations were performed using SPSS (Vers. 28.0, IBM, Armonk, USA). The parameter $\Delta\mathrm{RoM}_{\mathrm{SlowToFast}}$ was introduced to examine the resulting change in RoM between the conditions of slow walking and fast walking and to compare the sensitivity of the systems using a paired t-test. The individual cases of subjects with amputation were examined descriptively, showing the time-normalized joint angle curves of the three level walking trials.

## Results

The data of one subject and the ankle data of one subject had to be excluded due to unnoticed marker loss, and the 1.8 m/s condition of one subject was also excluded due to technical difficulties with the 2D-VB systems, as no report could be generated. Comparing sagittal joint angles of the lower extremity demonstrated both a horizontal and vertical misalignment between the two systems at 1.4 m/s treadmill walking (Fig 1).

   Hardly any differences were found between left and right leg sides gained with 3D-MB, whereas differences between the mean joint curves of leg sides became evident with 2D-VB. Although the average deviations at normal treadmill walking speed were less than ± 6° for all joints, the limits of agreement showed high variability (Fig 2). The Bland-Altman plots exhibit a systematic deviation of 1.6°, 3.9° and 5.6° for hip, knee and ankle, respectively, between both methods (Fig 2).

   The highest variations in the individually measured differences occurred for the ankle joint, where the limits of agreement covered a range of up to 20° (Fig 2) In general, the RoM values (Table 2) of the ankle and knee joints were significantly lower with 2D-VB than with 3D-MB across all conditions (p < 0.001) with the exception of 1 m/s treadmill walking speed (p = 0.06; p = 0.32).

   The maximum mean absolute RoM deviation was 6.6° ± 3.9° for the ankle joint during level walking (Table 2). Although the maximum mean absolute RoM differences of the hip joint were 4.4° ± 2.6° during level walking, there was no tendency

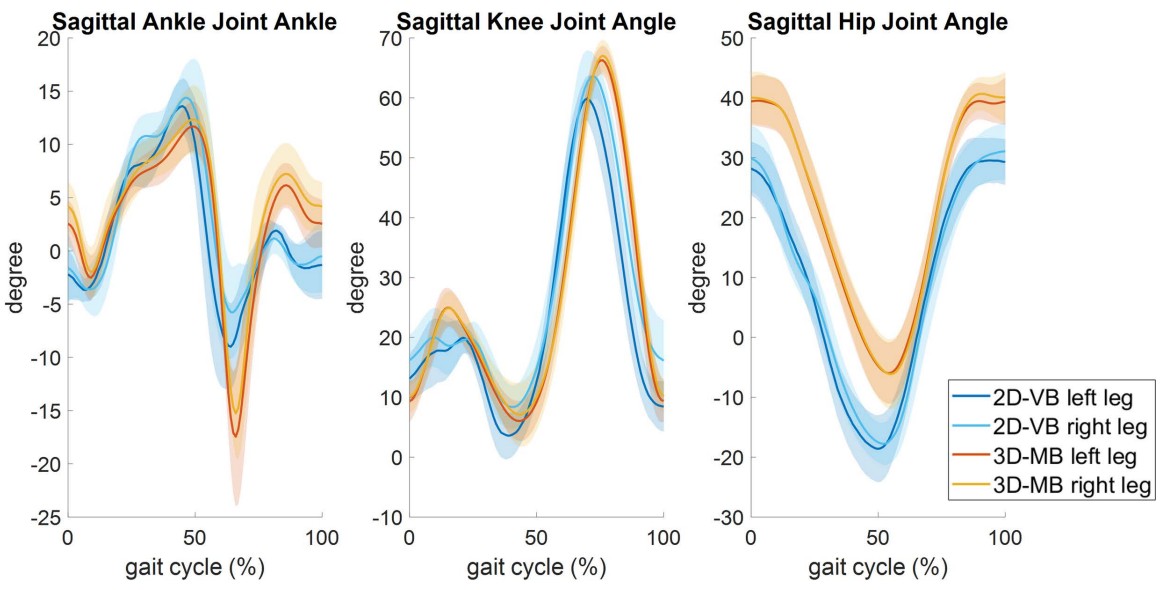

**Fig 1. Mean sagittal joint angle kinematics of healthy subjects during 1.4 m/s treadmill walking.**

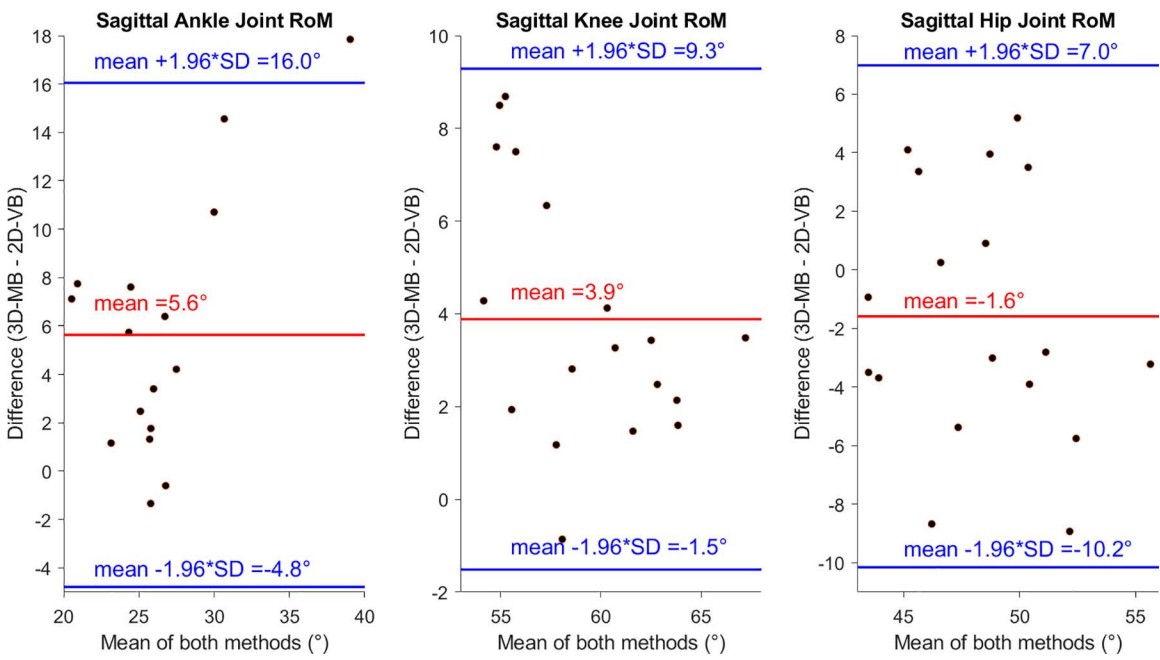

**Fig 2. RoM comparison of both systems at a walking speed of 1.4 m/s on the treadmill.**

**Table 2. Statistical RoM comparison of both systems during different walking conditions.**

| ankle joint RoM [deg] | 3D-MB | 2D-VB | p-value | r | ICC (3,1) |
|---|---|---|---|---|---|
| **1 m/s** | 25.4±6.5 | 22.8±3.6 | 0.06 | 0.67 | 0.56 |
| **1.4 m/s** | 29.2±6.3 | 23.6±3.4 | <0.01 | 0.54 | 0.45 |
| **1.8 m/s** | 28.7±4.5 | 24.2±3.7 | <0.01 | 0.49 | 0.48 |
| **level walking** | 31.4±7.4 | 25.0±5.6 | <0.01 | 0.82 | 0.79 |
| **knee joint RoM [deg]** | **3D-MB** | **2D-VB** | **p-value** | **r** | **ICC (3,1)** |
| **1 m/s** | 59.3±3.4 | 58.4±5.0 | 0.32 | 0.65 | 0.60 |
| **1.4 m/s** | 61.1±3.4 | 57.2±4.7 | <0.01 | 0.82 | 0.77 |
| **1.8 m/s** | 63.5±3.0 | 60.3±3.5 | <0.01 | 0.83 | 0.82 |
| **level walking** | 59.3±2.9 | 54.8±3.4 | <0.01 | 0.49 | 0.48 |
| **hip joint RoM [deg]** | **3D-MB** | **2D-VB** | **p-value** | **r** | **ICC (3,1)** |
| **1 m/s** | 40.9±3.1 | 43.8±4.8 | <0.01 | 0.82 | 0.75 |
| **1.4 m/s** | 47.5±3.7 | 49.1±4.3 | 0.14 | 0.42 | 0.41 |
| **1.8 m/s** | 54.6±2.8 | 57.4±3.6 | 0.01 | 0.36 | 0.35 |
| **level walking** | 43.9±3.9 | 47.8±4.8 | <0.01 | 0.72 | 0.70 |

towards over- or underestimation with one system (Fig 2). A high correlation between systems was observed for the ankle and hip joint during level walking, for the knee joint at 1.4 and 1.8 m/s, and for the hip joint at 1 m/s gait speed (Table 2). The highest agreement between the systems across all joints for both the correlation and the mean RoM deviations was at 1 m/s treadmill walking, but with small differences between the conditions (Table 2). The consistency between systems ranges from poor to good and varies between conditions and joints with most ICC values in the moderate consistency range (Table 2).

**Table 3. RoM change between walking at slow (1 m/s) and fast (1.8 m/s) gait speed on the treadmill.**

|  | 3D-MB | 2D-VB | p-value | r | ICC (3,1) |
|---|---|---|---|---|---|
| ankle joint ΔRoM$_{SlowToFast}$ [deg] | −3.3±3.8 | −1.4±3.7 | 0.15 | 0.12 | 0.12 |
| knee joint ΔRoM$_{SlowToFast}$ [deg] | −4.1±3.9 | −2.2±4.6 | 0.05 | 0.65 | 0.65 |
| hip joint ΔRoM$_{SlowToFast}$ [deg] | −13.4±2.2 | −13.2±4.7 | 0.87 | 0.65 | 0.49 |

**Table 4. Statistical comparison of time-distance parameter of both systems.**

| Speed [m/s] | 3D-MB | 2D-VB | p-value | r | ICC (3,1) |
|---|---|---|---|---|---|
| 1 m/s | 1.0 | 1.1±0.4 | <0.01 | ./. | ./. |
| 1.4 m/s | 1.4 | 1.5±0.1 | 0.02 | ./. | ./. |
| 1.8 m/s | 1.8 | 1.8±0.1 | 0.92 | ./. | ./. |
| level walking | 1.2±0.1 | 1.4±0.1 | <0.01 | 0.59 | 0.57 |
| **Cadence [steps/min]** | **3D-MB** | **2D-VB** | **p-value** | **r** | **ICC (3,1)** |
| 1 m/s | 103.0±4.8 | 102.9±5.4 | 0.95 | 0.85 | 0.84 |
| 1.4 m/s | 118.5±4.2 | 118.1±4.8 | 0.62 | 0.88 | 0.87 |
| 1.8 m/s | 134.2±4.3 | 131.2±3.2 | <0.01 | 0.88 | 0.85 |
| level walking | 111.6±5.3 | 115.5±5.3 | <0.01 | 0.93 | 0.93 |
| **Step length [m]** | **3D-MB** | **2D-VB** | **p-value** | **r** | **ICC (3,1)** |
| 1 m/s | 0.58±0.03 | 0.62±0.04 | 0.01 | 0.54 | 0.53 |
| 1.4 m/s | 0.71±0.03 | 0.74±0.04 | <0.01 | 0.55 | 0.52 |
| 1.8 m/s | 0.81±0.03 | 0.82±0.05 | 0.39 | 0.51 | 0.44 |
| level walking | 0.68±0.04 | 0.74±0.04 | <0.01 | 0.28 | 0.27 |

There were no significant differences regarding the individual RoM change from 1.0 to 1.8 m/s (ΔRoM$_{SlowToFast}$) condition between the systems (Table 3). However, the correlation was negligible ($r=0.12$) with comparatively large individual differences for the ankle joint.

The time-distance parameters differed between the systems for level walking ($p<0.001$; Table 4). Single parameters, in particular the walking speed at 1.8 m/s ($p=0.92$), the cadence at 1 m/s ($p=0.95$), and at 1.4 m/s ($p=0.62$), as well as the step length at 1.8 m/s ($p=0.39$), showed a high level of agreement for treadmill walking (Table 4). Good to excellent consistency is apparent for cadence in all conditions (Table 4).

For the subjects with leg amputations, the joint angle curves between both systems also showed an offset in terms of time and amplitude (Figs 3-4). In addition, the three trials captured with 2D-VB showed a higher variation. Of particular note are the differences in ankle plantarflexion on the prosthetic leg side in both the transtibial (Fig 3a) and transfemoral amputation (Fig 4a).

For the subject with the transfemoral amputation, higher differences in knee joint angles were observable between the systems (Fig 4b). Up to the mid-stance, a knee joint flexion up to 20° became evident with the 2D-VB, whereas this was measured far less with the 3D-MB system. In addition, the maximum knee joint flexion during the swing phase was reduced for the 2D-VB system. While only a reduced plantarflexion was measured with the 3D-MB system, it was significantly more pronounced with the 2D-VB, although also limited compared to the contralateral leg side (Figs 3a, 3d, 4a, 4d). For the contralateral leg side, the qualitative agreement of the joint motion measured with both systems, especially when considering the RoM, was comparable to the data of healthy subjects (Fig 1).

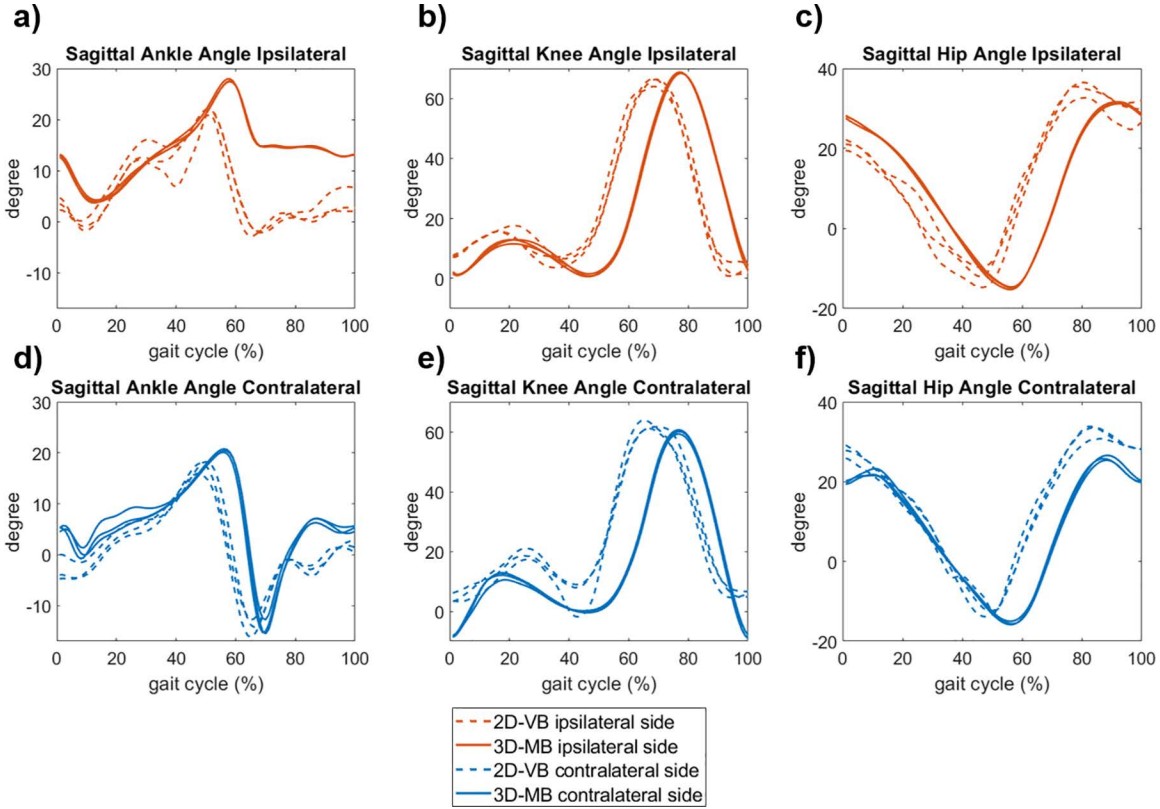

**Fig 3. Sagittal leg joint angles (three trials each) during level walking for the subject with a transtibial amputation.**

## Discussion

This study compares an emerging commercial markerless motion capture system within the orthopedic area with the gold standard of instrumented 3D motion analysis during different walking conditions. Therefore, this study focused on the clinically meaningful parameter of leg joint angle RoM. Although the RoM differs significantly between the systems, with only a few exceptions, the mean absolute deviation across joint angles and conditions is 4.3°. In accordance with the white-paper [19], the Bland-Altman plot reveals a notable spread of deviations. While individual joint angle RoM comparisons exhibit minor deviations, the 95% limits of agreement encompass a wide range up to 20°. Therefore, the accuracy is within the defined acceptable range of less than ± 5° for individuals. However, overall, the primary hypothesis that the knee and hip joint deviations are below the acceptable deviation of 5°, but at least below the value in the whitepaper (7.35°), cannot be fully confirmed. The mean deviations are in a similar range to those observed with a markerless multi-camera system (KinaTrax) [33]. In accordance with the system KinaTrax tested by Ripic et al. [33], the largest deviations relate to the ankle joint. However, precisely capturing ankle joint angles seems to be a major challenge across markerless systems [7]. With a mean deviation of 3.8°, the RoM deviation for the knee joint is higher than the values of <1° reported for other markerless systems [16,33], but below the manufacturer's own value of 7.35° [19]. The value is also below the minimal clinically important difference for sagittal knee angle RoM determined by Guzik et al. [34] for chronic stroke patients, with values of 8.5° and 6.8° for the affected and unaffected leg sides, respectively. For the hip joint, the deviation falls within the range of the minimal clinically important difference for stroke patients [35]. It can be confirmed that the results are better for the knee joint and hip joint than for the ankle joint, but these are also subject to a high degree of variability as shown by the limits of agreement.

**Fig 4. Sagittal leg joint angles (three trials each) during level walking for the subject with a transfemoral amputation.**

Often, the relationship between the measurements from different systems is of much greater importance than the absolute deviations. In contrast to some of the joint angle curves presented by Menychtas et al. [14], the curves from the presently tested 2D-VB system exhibit a qualitatively similar pattern to the 3D-MB data. However, as seen in another comparison of markerless tracking with computer vision techniques and 3D-MB [13], both a vertical and horizontal offset of the joint angle curves are observed. While a vertical offset is mainly caused by different segment and joint definitions between systems, the time-normalized joint angle curves of the 2D-VB system tested show different values at the beginning (0% gait cycle) and end (100%) of the gait cycle, leading to the suspicion of missing information. A sampling rate of 120 Hz is recommended for the precise determination of spatiotemporal parameters and gait events [36,37] and thus the temporal shift could be affected by a too low sampling frequency with the tablet. Nevertheless, the joint angles RoM values primarily show a moderate to high correlation between the systems across conditions and joints. However, except for the 1 m/s treadmill walking condition (mean r = 0.71), one joint angle at least shows a correlation of r < 0.5 in every condition. Therefore, these correlations are clearly lower than those for the knee joint angle RoM reported by Hu et al. [16] in a scientific setting and the correlation specified by the manufacturer for a simple squat movement [19].

The measured joint angle RoM change between slow treadmill walking (1 m/s) and fast treadmill walking (1.8 m/s) was determined and compared to examine the clinically meaningful reliability of individual changes across different conditions. No significant differences were found between systems, and a moderate correlation between the RoM changes was observed in the knee and hip joints. It can therefore be hypothesized that the system is capable of tracking multiple subjects with varying levels of accuracy, while reliably detecting changes within individuals.

A qualitative examination of individual cases with a lower limb amputation reveals that the 2D-VB system tested can detect joint angle differences between the prosthetic and contralateral leg side. However, the pattern displays a plantar-flexion at the transition from stance to swing phase in both subjects with an amputation, and a knee flexion in the stance phase in the subject with a transfemoral amputation, which is impossible due to the prosthetic fitting. Heitzmann et al. [38] report similar findings for markerless motion capture of subjects with a transtibial amputation with Theia3D and discuss insufficient training of the algorithms in the case of subjects with a prosthetic fitting. Therefore the algorithms should also be trained with other orthopedic devices so that they can be used successfully in prosthetic and orthotic care.

Although the markerless time-distance parameters of the healthy subjects differ significantly in some conditions from the marker-based system, the absolute differences between the systems are small and markedly less than the differences between the conditions. A recent review of markerless camera-based 3D motion capture systems [7] reports a high level of agreement for the time-distance parameters. Although differences are found in this study, the currently tested markerless system can be used to estimate these parameters.

One limitation of this pilot study is the small number of subjects and the lack of statistical analysis of subjects with an amputation. Furthermore, a young and healthy cohort was examined, so the generalizability of the results for other age groups and pathological gait patterns remains unclear. In particular, clinical populations in which abnormalities occur in planes other than the sagittal plane, such as cerebral palsy [39], should be investigated in the future. Given the soft-tissue artefacts commonly associated with measuring obese individuals using 3D-MB [40], a study involving obese participants and utilizing a markerless approach would be valuable as it could explore whether this methodology might provide advantages for motion analysis in this population. Since both measurement techniques in this study use surface information, resulting in a lack of data on the true movement of the bony structures, a co-registration with MRI data, as performed by Mihcin et al. [41], could be useful in answering this question. Although the sensitivity of the 2D-VB measuring RoM differences of individual subjects between two conditions is determined in this study, the reliability in the form of a between-days investigation should be examined to provide a complete statement for patient progress monitoring.

Furthermore, different versions of the 2D-VB system may have influenced data collection and processing. The manufacturer continuously updates the software and algorithms, so it can be assumed that recent versions yield improved results by enhancing both the volume and variability of the training dataset. The recent system report also offers many other parameters, such as a frontal plane analysis, which should be investigated in future studies. Taking into account the continuous progress and the emergence of new algorithms for gait pattern recognition [42], it can be assumed that the accuracy and application expansion will continue to develop rapidly. In this study, the temporal offset in comparison with 3D-MB was not examined in detail because it is not particularly relevant for many clinical purposes. A potential source of uncertainty is the manual numerical extraction of the report's joint angle graphs, which had no alternative due to the lack of the original data. However, Riaz [43] reported consistent results between numerical extracted and experimental data using the used tool. Nevertheless, future studies should refer to the raw data in consultation with the manufacturer. In addition, measurement-related influences such as illumination of the room, exact position and angle of the tablet and the tablet's camera should be investigated.

Data acquisition with the tablet and app is easy to handle and the report is created quickly. There were some technical difficulties in the 1.8 m/s condition, where measurements had to be repeated due to an unspecific error message. At present, the system appears to provide plausible patterns for lower limb joint angle kinematics, but the validity of the RoM is affected by high inter-subject variability. As described in a recent position paper [18], 2D-VB is not an equivalent replacement for 3D-MB but instead complements it and has the potential to be used in individual treatment monitoring or comparison of conditions within a subject. The handling of the 2D-VB system was simple in this study and was considerably more time-efficient than the instrumentation of the participants for the 3D-MB measurement. With the significantly lower price and the lack of need for professional staff, the system has the potential to be widely used in everyday clinical practice. Participants in this study were examined wearing shorts with a visible pelvis in a well-lit laboratory setting. This is

an idealized situation, so caution should be exercised in interpreting these results for the clinical setting. Therefore, future studies should investigate the system's accuracy in a more practice-oriented environment. In addition, the practicability and satisfaction of users should be investigated in future studies in everyday clinical practice using questionnaires. As this study indicates no differences in accuracy across walking speeds, future study protocols can limit their scope to single speeds.

## Conclusion

The system tested is a commercial tool that quickly and easily provides a complete biomechanical report of the lower extremity. The range of motion output differs from the gold standard of an instrumented 3D gait analysis and the accuracy is subject to high inter-subject variability. It therefore would be better suited for monitoring the progress of individuals than for cohort studies. Between-day repeatability needs to be investigated before the system can be integrated into daily clinical practice. In the future, the algorithms should also be trained with a broader range of clinical populations and subjects wearing orthopedic devices so that deviations from a healthy gait can be determined with greater sensitivity.

## Supporting information

**S1 Table. Raw data healthy subjects.** Raw data of all trials (M##) of the healthy subjects (0##) from the gait analysis recorded with the Orthelligent App (OV) and with the Qualisys system (Qual).
(ZIP)

**S2 Table. Raw data subjects with amputation.** Raw data of all trials (M##) of the subjects with amputation (0##) from the gait analysis recorded with the Orthelligent App (OV) and with the Qualisys system (Qual).
(ZIP)

## Acknowledgments

The authors would like to thank all the subjects for their patience and Henning Lauterbach for providing the tablet and the Orthelligent VISION app.

## Author contributions

**Conceptualization:** Frithjof Doerks, Eike Jakubowitz, Bastian Welke.

**Formal analysis:** Michael Schwarze.

**Investigation:** Fenna Harms.

**Software:** Frithjof Doerks, Fenna Harms.

**Supervision:** Eike Jakubowitz, Bastian Welke.

**Visualization:** Frithjof Doerks.

**Writing – original draft:** Frithjof Doerks.

**Writing – review & editing:** Fenna Harms, Michael Schwarze, Eike Jakubowitz, Bastian Welke.

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
