## [Decision Letter · Decision Letter 0]

7 Mar 2025

PONE-D-25-07386Gait assessment using a 2D video-based motion analysis app in healthy subjects and subjects with lower limb amputation – a pilot studyPLOS ONE

Dear Dr. Welke,

Thank you for submitting your manuscript to PLOS ONE. After careful consideration, we feel that it has merit but does not fully meet PLOS ONE’s publication criteria as it currently stands. Therefore, we invite you to submit a revised version of the manuscript that addresses the points raised during the review process.

We look forward to receiving your revised manuscript.

Kind regards,

Yaodong Gu

Academic Editor

PLOS ONE

2. We note you have included a table to which you do not refer in the text of your manuscript. Please ensure that you refer to Table 1-4 in your text; if accepted, production will need this reference to link the reader to the Table.

Additional Editor Comments (if provided):

Reviewers' comments:

Reviewer's Responses to Questions

**Comments to the Author**

1. Is the manuscript technically sound, and do the data support the conclusions?

Reviewer #1: Partly

Reviewer #2: Partly

2. Has the statistical analysis been performed appropriately and rigorously? 

Reviewer #1: No

Reviewer #2: Yes

3. Have the authors made all data underlying the findings in their manuscript fully available?

Reviewer #1: Yes

Reviewer #2: Yes

4. Is the manuscript presented in an intelligible fashion and written in standard English?

Reviewer #1: No

Reviewer #2: Yes

5. Review Comments to the Author

Reviewer #1: The study addresses a gap in clinical gait analysis by comparing a commercial 2D video-based motion analysis system to the gold standard 3D motion capture. The increasing use of markerless motion capture systems makes this study particularly relevant. The study presents well-defined hypotheses regarding the expected deviation of the 2D system compared to the 3D standard. The focus on intra-subject variability and the app’s potential for clinical applications is the research question. The inclusion of multiple treadmill speeds and overground walking provides a comprehensive test environment. However, in the field of biomechanics, a proof of evidence is required starting with in vitro studies. Instead of starting with healthy subjects, they could have started with fully restrained system which operates in 2 D and then compare their results with a 3D system, to provide systematic error information.

Please refer to the study Simultaneous validation of wearable motion capture system for lower body applications: over single plane range of motion (ROM) and gait activities, BIOMEDICAL ENGINEERING-BIOMEDIZINISCHE TECHNIK, 2022, 0013-5585, 67, 3, 185-199. And explain why you did not start with a fully restrained system by citing this paper and explain your error margins in each plane before moving on to healthy subjects.

Also explain the reasons for your thread mill speeds. Why in particular you focus on those speeds ? And why did you choose 200 fps ? Would your results be different if was 100 Fps or 300 Fps. Do you have any evidence in your fps selection and walking speeds? You may refer to the study "Cut-off Frequency Estimation Methods for Biomechanical Data Filtering," 2018 Medical Technologies National Congress (TIPTEKNO), Magusa, Cyprus, 2018, pp. 1-4, doi: 10.1109/TIPTEKNO.2018.8596916. for providing information and explain your limitations by citing this study.

You did not explain your Bland Altman graphs properly. For example the deviations of the mean from the zero shows systematic bias so you have a high bias for ankle and low bias for hip. Also the distance between upper and lower limits shows low repeatability. The data points following out of the upper and lower limits correspond to low p values whereas no outliers between the upper and lower limits correspons to high p values in students T test. You should place this in the discussions providing a validation of your statistics. There is no point in using pearson value as it only relates to the linearity between the two systems.

Also you might have made a co-registration between the two systems to be able to make comparisons as a start of your proof of concept and eliminate the markers afterwards as in the following paper. Please cite this paper and explain it as a limitation of your study. Methodology on Co-registration of MRI and Optoelectronic Motion Capture Marker Sets: In-vivo Wrist Case Study, Hitit Journal of Science and Engineering, Available online in 01/06/2019.

Limitations & Areas for Improvement:

1. Sample Size & Generalizability:The study only includes 10 healthy subjects and 2 subjects with lower limb amputation. While it is labeled as a pilot study, the findings cannot be generalized to a larger population, particularly for individuals with gait impairments. Diverse range of patients, including older adults and individuals with various orthopedic conditions could be added.

2. Variability & System Limitations:The results show high inter-subject variability, making it difficult to draw firm conclusions on the reliability of the 2D system for cohort studies. Differences between the prosthetic and sound limbs in amputees were detected, but some joint movements (e.g., stance-phase knee flexion in the transfemoral amputee) were inaccurately represented, likely due to algorithmic limitations in detecting prosthetic movement.

3. Lack of Between-Day Reliability Testing:

The study does not assess the repeatability of the 2D system across different sessions.

Clinical applications would benefit from understanding whether the system produces consistent results across multiple trials on different days.

4. Accuracy of Data Extraction:

The WebPlotDigitizer tool was used to extract numerical values from graphs.This introduces potential digitization errors, and a direct data export function from the 2D-VB app would improve accuracy. If possible, future work should advocate for the manufacturer to provide raw numerical output.

5. Temporal Misalignment & Missing Gait Data: There are horizontal and vertical offsets in the 2D kinematic curves compared to the 3D system.The study suggests that data may be missing at the start and end of the gait cycle, raising concerns about whether the system captures the full gait cycle accurately. Think of referring to the co registration between the systems and refer to the paper ‘Methodology on Co-registration of MRI and Optoelectronic Motion Capture Marker Sets: In-vivo Wrist Case Study, Hitit Journal of Science and Engineering, Available online in 01/06/2019. ‘

6. Limited Discussion on Clinical Feasibility: While the study suggests potential clinical applications, it does not address how clinicians would integrate this tool into real-world settings. Future work should explore factors like time efficiency, user-friendliness, and cost-effectiveness compared to 3D marker-based systems.

1. Include Between-Day Reliability Testing:

o A longitudinal study design would help determine whether the 2D system provides consistent measurements over time.

2. Improve Data Extraction Methods:

o Work with the manufacturer to gain direct access to raw numerical data instead of relying on digitized graphs.

3. Investigate Real-World Feasibility:

o Conduct usability studies to assess whether clinicians and patients find the app practical for daily use.

Conclusion:

This study provides initial insights into the feasibility of using a commercial 2D video-based gait analysis app as an alternative to traditional 3D motion capture. While the results support its potential for intra-subject monitoring, high variability limits its application for inter-subject comparisons. The sample size is not enough to make wide impactful conclusions. They should include improved reliability assessments and explore real-world usability in clinical practice.

Reviewer #2: The article titled "Gait assessment using a 2D video-based motion analysis app in healthy subjects and subjects with lower limb amputation – a pilot study" evaluates the accuracy of a 2D video analysis application (Orthelligent VISION) in gait analysis for both healthy subjects and those with lower limb amputations by comparing it to a 3D optical motion capture system. It validates the app's potential for intra-subject progress monitoring but identifies high inter-subject variability as a possible limitation for its clinical application. However, there are several issues within the manuscript that affect its overall quality, detailed as follows:

1. The introduction does not adequately cover the key advancements in 2D video analysis in biomechanics over recent years, particularly research on algorithm optimization incorporating deep learning or transfer learning. Additionally, there is a lack of comparative analysis with existing commercial tools (such as Deep LabCut and Theia3D), which fails to highlight the unique advantages of Orthelligent VISION. It is recommended to supplement this information.

2. While this study claims to explore the accuracy of 2D video analysis, it does not sufficiently explain the system's practical value in clinical applications. For example, whether it can serve as an alternative to 3D systems or is only suitable for screening and individual monitoring? This section could be expanded to clarify the clinical or practical application background of the study.

3. The study includes only 10 healthy subjects and 2 amputees, making the sample size too small to generalize to larger populations. Moreover, it does not specify whether the sample size was determined through statistical calculations. It is recommended to increase the sample size or provide a basis for the sample size calculation.

4. The results section mainly uses Bland-Altman analysis and t-tests but does not provide measures of system measurement error (such as root mean square error RMSE) or ICC (intraclass correlation coefficient) to assess measurement consistency. It is suggested to add ICC or Lin’s concordance correlation coefficient to offer a more comprehensive evaluation of measurement consistency.

5. The study mentions only the small sample size but does not discuss other limitations, such as the impact of camera angles on measurement accuracy and the applicability of 2D systems in non-standard gaits. It is recommended to supplement suggestions for improving experimental methods, data analysis, and system algorithms to make the discussion more complete.

6. The study explores the feasibility of a 2D video-based motion capture system for gait analysis but faces challenges such as measurement inaccuracies, high inter-subject variability, and limitations in detecting key gait events. While traditional 3D marker-based systems are the gold standard, their high cost and complexity limit their use, driving interest in 2D video-based alternatives. However, these systems often struggle with accuracy and feature extraction. Recent studies have shown the potential of machine learning in improving gait analysis. For instance, (A new method proposed for realizing human gait pattern recognition. https://doi.org/10.1016/j.gaitpost.2023.10.019) introduced a novel recognition approach with applications in sports and clinical settings, while (Explaining the differences of gait patterns between high and low‑mileage runners with machine learning. https://doi.org/10.1038/s41598-022-07054-1) demonstrated how machine learning can uncover subtle gait differences. Therefore, integrating machine learning algorithms into 2D video-based systems could enhance their accuracy in detecting joint kinematics and key gait events, addressing current limitations in feature extraction and inter-subject variability. Additionally, further research should focus on training these algorithms with diverse populations, including individuals with gait abnormalities, to improve the system's applicability in clinical and rehabilitative settings.

6. PLOS authors have the option to publish the peer review history of their article (what does this mean? ). If published, this will include your full peer review and any attached files.

**Do you want your identity to be public for this peer review?** For information about this choice, including consent withdrawal, please see our Privacy Policy .

Reviewer #1: No

Reviewer #2: No

---

## [Author Response · Author response to Decision Letter 1]

25 Apr 2025

Dear Dr. Yaodong Gu

Thanks a lot for the opportunity to revise our paper in order to resubmit it to PLOS ONE. We have worked on all comments and have implemented all of the suggestions and points of criticism. The present response is done in a stepwise way regarding the reviewer’s comments (italic). Corresponding changes and additions have been highlighted as track changes in the “manuscript with track changes”.

Reviewer #1: Reviewer #1: The study addresses a gap in clinical gait analysis by comparing a commercial 2D video-based motion analysis system to the gold standard 3D motion capture. The increasing use of markerless motion capture systems makes this study particularly relevant. The study presents well-defined hypotheses regarding the expected deviation of the 2D system compared to the 3D standard. The focus on intra-subject variability and the app’s potential for clinical applications is the research question. The inclusion of multiple treadmill speeds and overground walking provides a comprehensive test environment. However, in the field of biomechanics, a proof of evidence is required starting with in vitro studies. Instead of starting with healthy subjects, they could have started with fully restrained system which operates in 2 D and then compare their results with a 3D system, to provide systematic error information.

Please refer to the study Simultaneous validation of wearable motion capture system for lower body applications: over single plane range of motion (ROM) and gait activities, BIOMEDICAL ENGINEERING-BIOMEDIZINISCHE TECHNIK, 2022, 0013-5585, 67, 3, 185-199. And explain why you did not start with a fully restrained system by citing this paper and explain your error margins in each plane before moving on to healthy subjects.

Many thanks for this advice and the relevant literature. We also believe that testing in fully restricted conditions and in-vitro is absolutely desirable. However, as we are not the manufacturers of the tool and can therefore only limit ourselves to the existing functions of the tool, in-vitro testing is unfortunately not possible. The computer vision-based algorithm is trained to recognize the human shape while walking. Furthermore, at the time of the study, only observation in the sagittal plane was possible. Nevertheless, we have made an addition including a citation in the introduction to explain the correct methodological procedure (line 99-101).

Also explain the reasons for your thread mill speeds. Why in particular you focus on those speeds ? And why did you choose 200 fps ? Would your results be different if was 100 Fps or 300 Fps. Do you have any evidence in your fps selection and walking speeds? You may refer to the study "Cut-off Frequency Estimation Methods for Biomechanical Data Filtering," 2018 Medical Technologies National Congress (TIPTEKNO), Magusa, Cyprus, 2018, pp. 1-4, doi: 10.1109/TIPTEKNO.2018.8596916. for providing information and explain your limitations by citing this study.

Thank you for the question. The choice of treadmill speeds is explained in lines 136 to 137. The sampling rate of 200 Hz corresponds to the standard setting for the gait analysis studies in our laboratory. This is above the frequency of 35 Hz recommended by Allseits et al. [1]. Thanks again for the comment, by looking at the literature on sampling rate we have found another valuable point for our discussion. In other studies, a frequency of 120 Hz is recommended for the calculation of spatio-temporal parameters and the precise determination of gait events [2,3]. It is possible that the lower recording frequency with the tablet (Ipad) affects the temporal offset between the systems in our study. We have addressed this point in lines 269-272.

You did not explain your Bland Altman graphs properly. For example the deviations of the mean from the zero shows systematic bias so you have a high bias for ankle and low bias for hip. Also the distance between upper and lower limits shows low repeatability. The data points following out of the upper and lower limits correspond to low p values whereas no outliers between the upper and lower limits correspons to high p values in students T test. You should place this in the discussions providing a validation of your statistics. There is no point in using pearson value as it only relates to the linearity between the two systems.

We have added some more detailed explanation to the Bland-Altman plots regarding the bias (lines 185-186). As discussed in a recent systematic review comparing different motion capture systems [4], the pearson coefficient is used to determine the relative agreement.

Also you might have made a co-registration between the two systems to be able to make comparisons as a start of your proof of concept and eliminate the markers afterwards as in the following paper. Please cite this paper and explain it as a limitation of your study. Methodology on Co-registration of MRI and Optoelectronic Motion Capture Marker Sets: In-vivo Wrist Case Study, Hitit Journal of Science and Engineering, Available online in 01/06/2019.

Unlike in the study recommended above, both procedures in this study are surface measurements, so that they could be recorded synchronously. One limitation, however, is the lack of correlation with the movement of the bony structures. One method uses the morphology of the body and the other the marker points as the basis for calculating the kinematics. Co-registration of both systems with an MRI system would be desirable, but not possible within the scope of this study. Nevertheless, we have included this point in the limitations (lines 307-309).

Limitations & Areas for Improvement:

1. Sample Size & Generalizability:The study only includes 10 healthy subjects and 2 subjects with lower limb amputation. While it is labeled as a pilot study, the findings cannot be generalized to a larger population, particularly for individuals with gait impairments. Diverse range of patients, including older adults and individuals with various orthopedic conditions could be added.

Thank you for this important comment. Since no information on the accuracy of the system was available at the time the study was conducted, it is difficult to carry out valid sample size planning. The observation of statistically significant differences suggests that the sample size was sufficient to detect the effect observed in this study. In future studies, we plan to expand the cohort to include a more diverse group of participants, especially individuals with various orthopedic and neurological gait impairments. This will allow us to further validate our findings and explore their applicability across a wider clinical spectrum.

2. Variability & System Limitations:The results show high inter-subject variability, making it difficult to draw firm conclusions on the reliability of the 2D system for cohort studies. Differences between the prosthetic and sound limbs in amputees were detected, but some joint movements (e.g., stance-phase knee flexion in the transfemoral amputee) were inaccurately represented, likely due to algorithmic limitations in detecting prosthetic movement.

That is correct and that is how we address it in the discussion section. As we are not the manufacturers of the system, we cannot improve the algorithms, we can only point out current limitations.

3. Lack of Between-Day Reliability Testing:

The study does not assess the repeatability of the 2D system across different sessions.

Clinical applications would benefit from understanding whether the system produces consistent results across multiple trials on different days.

We fully agree with this. That is why we have already addressed this in our discussion in lines 310-312. A between-day study should be conducted in the future. However, from our perspective, identifying differences between two conditions within a single subject, as done in this study, also has high clinical relevance.

4. Accuracy of Data Extraction:

The WebPlotDigitizer tool was used to extract numerical values from graphs.This introduces potential digitization errors, and a direct data export function from the 2D-VB app would improve accuracy. If possible, future work should advocate for the manufacturer to provide raw numerical output.

That's a good hint. As a research laboratory, we wanted to investigate the system as neutrally as possible, without prior contact with the manufacturer. In line 323, we explain the possible limitation of manual data extraction and refer to literature that emphasizes high accuracy. We have now also added a sentence to the discussion to give an outlook on future studies (lines 324-325).

5. Temporal Misalignment & Missing Gait Data: There are horizontal and vertical offsets in the 2D kinematic curves compared to the 3D system.The study suggests that data may be missing at the start and end of the gait cycle, raising concerns about whether the system captures the full gait cycle accurately. Think of referring to the co registration between the systems and refer to the paper ‘Methodology on Co-registration of MRI and Optoelectronic Motion Capture Marker Sets: In-vivo Wrist Case Study, Hitit Journal of Science and Engineering, Available online in 01/06/2019.

The concerns about whether the 2D video-based system captures the complete gait cycle are plausible. We have added a point on possibly insufficient sampling frequency with a tablet as well as with the recommended paper (please find the section above).

6. Limited Discussion on Clinical Feasibility: While the study suggests potential clinical applications, it does not address how clinicians would integrate this tool into real-world settings. Future work should explore factors like time efficiency, user-friendliness, and cost-effectiveness compared to 3D marker-based systems.

That's a good hint. As an independent research laboratory, we wanted to be as neutral as possible. But of course it makes sense to include the advantages of the system in the discussion. We have added these aspects to lines 333-3336 & lines 339-341.

1. Include Between-Day Reliability Testing:

o A longitudinal study design would help determine whether the 2D system provides consistent measurements over time.

Thank you for this important comment. We will take into account the influence of different time points on the measurements with the application in a future study.

2. Improve Data Extraction Methods:

o Work with the manufacturer to gain direct access to raw numerical data instead of relying on digitized graphs.

For future studies, we will contact the manufacturer of the application and discuss the possibility of exporting the raw data.

3. Investigate Real-World Feasibility:

o Conduct usability studies to assess whether clinicians and patients find the app practical for daily use.

The original idea for this study actually came from our neighboring medical supply store (orthopedic technology). The staff there would like to use the app in their daily work with their patients. They are also likely to be very interested in being able to perform simpler movement analyses. We also know from our clinical colleagues in orthopedics that they would like to have this option. Therefore, it is conceivable that we will conduct surveys of users and patients with questionnaires in further work.

Conclusion:

This study provides initial insights into the feasibility of using a commercial 2D video-based gait analysis app as an alternative to traditional 3D motion capture. While the results support its potential for intra-subject monitoring, high variability limits its application for inter-subject comparisons. The sample size is not enough to make wide impactful conclusions. They should include improved reliability assessments and explore real-world usability in clinical practice.

Reviewer #2: The article titled "Gait assessment using a 2D video-based motion analysis app in healthy subjects and subjects with lower limb amputation – a pilot study" evaluates the accuracy of a 2D video analysis application (Orthelligent VISION) in gait analysis for both healthy subjects and those with lower limb amputations by comparing it to a 3D optical motion capture system. It validates the app's potential for intra-subject progress monitoring but identifies high inter-subject variability as a possible limitation for its clinical application. However, there are several issues within the manuscript that affect its overall quality, detailed as follows:

1. The introduction does not adequately cover the key advancements in 2D video analysis in biomechanics over recent years, particularly research on algorithm optimization incorporating deep learning or transfer learning. Additionally, there is a lack of comparative analysis with existing commercial tools (such as Deep LabCut and Theia3D), which fails to highlight the unique advantages of Orthelligent VISION. It is recommended to supplement this information.

Many thanks for this helpful hint. We have consulted additional literature (D’Souza et al., 2024; Roggio et al., 2024; Tang et al., 2024; Tien et al., 2022) and added the differentiation from related solutions such as DeepLabCut and Theia3D. Please find the changes in lines 52-54, 56-60 and 82-83.

2. While this study claims to explore the accuracy of 2D video analysis, it does not sufficiently explain the system's practical value in clinical applications. For example, whether it can serve as an alternative to 3D systems or is only suitable for screening and individual monitoring? This section could be expanded to clarify the clinical or practical application background of the study.

As a research laboratory with no connection to the manufacturer of the 2D video system, we wanted to be as independent as possible. Nevertheless, your comment in point 1 and the addition of literature from us serves to further emphasize the clinical value. Furthermore, we have emphasized in line 58 that this tool provides a possibility for objective gait analysis that can influence the clinical decision-making process. In addition, we have added in line 86 that no laboratory is required and that it is therefore available to a wide range of users.

3. The study includes only 10 healthy subjects and 2 amputees, making the sample size too small to generalize to larger populations. Moreover, it does not specify whether the sample size was determined through statistical calculations. It is recommended to increase the sample size or provide a basis for the sample size calculation.

Thank you for this important comment. Since no information on the accuracy of the system was available at the time the study was conducted, it is difficult to carry out valid sample size planning. The observation of statistically significant differences suggests that the sample size was sufficient to detect the effect observed in this study. In future studies, we plan to expand the cohort to include a more diverse group of participants, especially individuals with various orthopedic and neurological gait impairments. This will allow us to further validate our findings and explore their applicability across a wider clinical spectrum.

4. The results section mainly uses Bland-Altman analysis and t-tests but does not provide measures of system measurement error (such as root mean square error RMSE) or ICC (intraclass correlation coefficient) to assess measurement consistency. It is suggested to add ICC or Lin’s concordance correlation coefficient to offer a more comprehensive evaluation of measurement consistency.

Thanks for the advice, this is certainly a good addition. We have added the ICC for determining consistency in Tables 2-4. The definition was also supplemented in the statistics section (lines 165-168). Furthermore, we added the information in the results section (lines 204-206; lines 216-217).

5. The study mentions only the small sample size but does not discuss other limitations, such as the impact of camera angles on measurement accuracy and the applicability of 2D systems in non-standard gaits. It is recommended to supplement suggestions for improving experimental methods, data analysis, and system algorithms to make the discussion more complete.

Thank you for your comment, which we fully agree with. We have therefore reemphasized the need to stud

---

## [Decision Letter · Decision Letter 1]

28 Apr 2025

Gait assessment using a 2D video-based motion analysis app in healthy subjects and subjects with lower limb amputation – a pilot study

PONE-D-25-07386R1

Dear Dr. Welke,

We’re pleased to inform you that your manuscript has been judged scientifically suitable for publication and will be formally accepted for publication once it meets all outstanding technical requirements.

Kind regards,

Yaodong Gu

Academic Editor

PLOS ONE

Additional Editor Comments (optional):

Reviewers' comments:

Reviewer's Responses to Questions

**Comments to the Author**

1. If the authors have adequately addressed your comments raised in a previous round of review and you feel that this manuscript is now acceptable for publication, you may indicate that here to bypass the “Comments to the Author” section, enter your conflict of interest statement in the “Confidential to Editor” section, and submit your "Accept" recommendation.

Reviewer #1: All comments have been addressed

Reviewer #2: (No Response)

2. Is the manuscript technically sound, and do the data support the conclusions?

Reviewer #1: Yes

Reviewer #2: Yes

3. Has the statistical analysis been performed appropriately and rigorously? 

Reviewer #1: Yes

Reviewer #2: Yes

4. Have the authors made all data underlying the findings in their manuscript fully available?

Reviewer #1: Yes

Reviewer #2: Yes

5. Is the manuscript presented in an intelligible fashion and written in standard English?

Reviewer #1: Yes

Reviewer #2: Yes

6. Review Comments to the Author

Reviewer #1: Thanks for making the requested changes, I have no further comments. You have addressed the requested changes properly. The paper seems to have improved alot in terms of quality now.

Reviewer #2: The author carefully revised the first review comments, supplemented some literature reviews, improved the methodological details, and expanded the statistical analysis and discussion contents to a certain extent. Overall, there has been an improvement. However, from the perspectives of scientificity and logical rigor, the manuscript still has certain problems, including insufficiently in-depth theoretical review, incomplete description of method details, insufficient analysis and explanation of results, and the discussion and outlook sections are still rather general. Overall, the manuscript still has a certain gap from meeting the standard for official publication. The detailed opinions are as follows:

1. The comparison of existing technologies is still somewhat weak. Although the author has added the introduction of tools such as DeepLabCut and Theia3D, the application of their algorithm optimizations (such as deep learning or transfer learning) in improving the accuracy of 2D motion capture has not been analyzed in detail. It is only briefly mentioned, which is not in-depth enough. It is suggested to briefly introduce the specific research on 2D systems through transfer learning and deep neural network optimization (not only listing references, but also 1-2 specific sentences summarizing the application effects).

2. The positioning of application scenarios is not clear enough. Although the description of "clinical application potential" has been added, it is still not clearly distinguished: Is this system used for preliminary screening, individual monitoring, or can it partially replace certain 3D-MB applications? Please clearly define the application positioning. Therefore, it is suggested to add a small paragraph to clarify that this system is mainly suitable for "clinical screening and monitoring of individual patients", rather than large-scale clinical studies or assessment of complex gait abnormalities.

3. The sample size and randomness of the subjects are insufficient. Although the reviewer's question about the small sample size was responded to, it was still not mentioned whether there was any gender or age bias (for example, 8 women and 2 men, would it affect the result?). Please supplement the description of the gender ratio deviation of the subjects and whether it may affect the RoM deviation.

4. 2D system data extraction method has the risk of deviation. Although the use of WebPlotDigitizer has been mentioned at present, there is no form of cross-validation, such as whether repeated extraction has been carried out to detect errors? In the "Data Analysis" section, briefly supplement and explain whether a review was conducted during data extraction to reduce manual extraction errors.

5. The statistical indicator lacks RMSE (Root Mean Square Error). Although ICC was added, RMSE or standard deviation size was not reported to better describe the level of systematic error. It is also worth noting that the interpretation of the charts is still not perfect. For example, the Bland-Altman plot has supplemented the description of bias, but has not explicitly pointed out: What does the width of the 95% consistent limit mean for clinical practical application?

6. There is a lack of out-of-sample extensibility prompts. Only the results of healthy individuals and two amputees are presented, and the latter is descriptive with no statistical indicators. Therefore, when presenting the data of amputees, please add a sentence stating: "Due to the extremely small sample size of the case study, the results are only for trend observation and need to be verified in a larger sample in the future."

7. The analysis of some limitations is not detailed enough. Although the sampling rate and equipment limitations were mentioned, the potential impacts of lighting, background interference, and camera angles on the stability of the system were not elaborated. The outlook for future applications is too general. Although it was mentioned that it is hoped to combine machine learning to improve the algorithm, there are no specific ideas, such as: Is it necessary to introduce multi-angle shooting? Or combined with deep neural networks? At the same time, it is suggested to add a summary to explain the possible differences in learning curves and adaptations for users with different professional backgrounds (doctors, rehabilitation technicians) in actual operations.

7. PLOS authors have the option to publish the peer review history of their article (what does this mean? ). If published, this will include your full peer review and any attached files.

**Do you want your identity to be public for this peer review?** For information about this choice, including consent withdrawal, please see our Privacy Policy .

Reviewer #1: No

Reviewer #2: No

---

## [Editor Report · Acceptance letter]

PONE-D-25-07386R1

PLOS ONE

Dear Dr. Welke,

I'm pleased to inform you that your manuscript has been deemed suitable for publication in PLOS ONE. Congratulations! Your manuscript is now being handed over to our production team.

Kind regards,

on behalf of

Professor Yaodong Gu

Academic Editor

PLOS ONE